# Characteristics influencing COVID-19 testing and vaccination among Spanish-speaking Latine persons in North Carolina

**Sandy K. Aguilar-Palma**[1]*, **Thomas P. McCoy**[2], **Lilli Mann-Jackson**[1], **Jorge Alonzo**[1], **Mohammed Sheikh Eldin Jibriel**[3], **Dorcas Mabiala Johnson**[3], **Tony Locklear**[3], **Amanda E. Tanner**[3], **Mark A. Hall**[1,4], **Alain G. Bertoni**[5], **Ana D. Sucaldito**[1], **Laurie P. Russell**[6], **Scott D. Rhodes**[1]

1 Department of Social Sciences and Health Policy, Wake Forest University School of Medicine, Winston-Salem, NC, United States of America, 2 School of Nursing, University of North Carolina Greensboro, Greensboro, NC, United States of America, 3 Department of Public Health Education, University of North Carolina Greensboro, Greensboro, NC, United States of America, 4 Wake Forest University School of Law, Winston-Salem, NC, United States of America, 5 Division of Public Health Sciences, Wake Forest University School of Medicine, Winston-Salem, NC, United States of America, 6 Department of Biostatistics and Data Science, Wake Forest University School of Medicine, Winston-Salem, NC, United States of America

* saguilar@wakehealth.edu

## Abstract

### Background

Latine populations in the United States continue to be disproportionately affected by COVID-19 with high rates of infection and mortality. Our community-based participatory research partnership examined factors associated with COVID-19 testing and vaccination within a particularly hidden, underserved, and vulnerable population: Spanish-speaking Latines.

### Methods

In 2023, native Spanish-speaking Latine interviewers conducted phone-based structured individual assessments with 180 Spanish-speaking, predominantly immigrant Latines across North Carolina. We used univariate and multivariable logistic regression analyses to examine associations between participant characteristics and COVID-19 testing and vaccination.

### Results

Participant mean age was 41.7 (*SD* = 13), and 77.2% of the sample reported being cisgender women. Most participants reported immigrating from Latin American countries (89.9%), being uninsured (66.1%), and lacking US immigration documentation (51.1%). While most reported ever being COVID-19 tested (83.3%) and ever being vaccinated against COVID-19 (84.4%), only 24% were up to date with vaccination. Nearly half of the sample reported one or more barriers to COVID-19 testing, and over one-quarter reported one or more barriers to COVID-19 vaccination. Higher educational attainment was significantly associated

**Data Availability Statement:** "Ethical restrictions apply to data access because the dataset contains

highly sensitive and potentially identifying information about a vulnerable population, Spanish-speaking immigrants. Potential researchers can request data from this study at: https://www.ncbi.nlm.nih.gov/projects/gap/cgi-bin/study.cgi?study_id=phs002946.v1.p1. The website has requesting information, data dictionaries, and data. RADx-UP through NIH may provide access to the dataset after assessing the purpose of the data use."

**Funding:** This work is supported by the US National Institute of Minority Health and Health Disparities [https://www.nimhd.nih.gov/; U01MD017431; PIs: SDR and AET]. The funders had no role in study design, data collection and analysis, decision to publish, or preparation of the manuscript.

**Competing interests:** The authors have declared that no competing interests exist.

with ever being tested ($P$ = .031). Fewer concerns about the vaccine, including fewer worries about side effects and having more confidence in vaccine effectiveness and safety, was associated with ever being vaccinated ($P$ < .001).

## Conclusions

Spanish-speaking Latines face barriers to getting tested and vaccinated against COVID-19. Although ever testing and ever vaccination rates were high, being up to date with recommended vaccinations was low. Educational attainment and concerns about the vaccine were associated with COVID-19 testing and vaccination, respectively. Our findings suggest the need for culturally congruent strategies to address the challenges facing Spanish-speaking Latines in the United States.

## Introduction

### COVID-19 and Latine communities in the United States

Racial and ethnic minorities in the United States continue to carry disproportionate rates of COVID-19 infection and mortality compared to their non-Latine White counterparts, and this is particularly true of US Latine communities [1]. Despite comprising 19.1% of the US population [2], since the beginning of the pandemic, Latines comprised 24.1% of reported COVID-19 cases [3]. Additionally, Latines in the United States also have experienced disproportionately high rates of hospitalization due to COVID-19, and COVID-19-related mortality remains higher among Latines [4]. Throughout the pandemic and continuing today, these disparities have been notably more pronounced among Spanish-speaking Latines [5–7].

### Barriers to COVID-19 testing and vaccination among Latines

Complex factors continue to drive low rates of COVID-19 testing and vaccination within Latine communities in the United States [1, 7, 8]. These factors include poverty, limited resources, and institutional and structural barriers (e.g., discrimination and limited transportation). Latines may have limited access to culturally congruent and updated information about COVID-19 testing and vaccination, including eligibility and how to access these services; believe they are not at risk for COVID-19; worry about missing work due to a positive test result or vaccine side effects; and assume that they must pay for a test or vaccine. Other challenges include fear of engaging with government systems (including public health) and concerns about the long-term health implications of testing and vaccination. These barriers are even greater among those who do not speak English [1, 7–12].

Previous research has also suggested that undocumented persons and those living in mixed-immigration status households may face more significant challenges in accessing health services, resulting in comparatively poorer health outcomes compared to their documented counterparts [8–12]. For many Latine persons, factors, such as the fear of deportation or the deportation of family members, uncertainty about required documentation, a desire to limit possible interactions with law enforcement, and racial profiling, have been linked to reduced use of health services [1, 13, 14].

Our community-based participatory research (CBPR) partnership's preliminary research identified additional barriers to COVID-19 testing and vaccination, including misinformation about testing and vaccination communicated through Spanish-language TV and social media

platforms, as well as through word-of-mouth within social networks; medical mistrust; and distrust in the test or vaccine itself (e.g., not believing that they will get the real vaccine as has been reported in Latin America [7, 11, 15–17]. Participants in our preliminary research also reported that many Latines do not trust, feel discouraged from using, and encounter difficulty accessing and navigating the US healthcare system, particularly non-English speakers and those less familiar with how to obtain healthcare resources and services in the United States [7].

Furthermore, it is well documented that some Latines fear using public health services given experiences with racial/ethnic discrimination, anti-immigration rhetoric, and policies and laws that discourage or restrict access [7, 13, 18–20]. Moreover, participants highlighted concern regarding some immigration policies, such as Section 287(g) of the Immigration and Nationality Act and the Secure Communities program [13], sanctuary city status [14, 21], and "public charge" (term used by US immigration officials and the media for individuals considered likely to become dependent on the government for subsistence). These policies may contribute to a reduced perception of access to and a disincentive for using COVID-related services. Many Latines are concerned that seeking these services might expose them or their family members to immigration enforcement actions or lead to adverse legal consequences related to their immigration status [1, 7].

Our CBPR partnership sought to better understand COVID-19 testing and vaccination among Spanish-speaking Latine persons living in North Carolina. While understanding of testing and vaccination is expanding, most studies do not focus on this population (i.e., Spanish-speaking and immigrant Latine persons in the US South) that tends to be hidden, underserved, and vulnerable.

## Methods

### Community-based participatory research

This analysis is part of a larger parent intervention study to assess the impact of an intervention harnessing natural helping through community-based peer navigators (known as Navegantes) and mHealth to increase COVID-19 testing and vaccination among Spanish-speaking Latine in North Carolina [1]. The parent study and this analysis were conceived and developed by our long-standing CBPR partnership, consisting of members from academic research institutions and representatives of Latine community organizations in North Carolina [1]. Additionally, the parent study is part of the National Institutes of Health (NIH) Rapid Acceleration of Diagnostics Underserved Populations (RADx-UP) initiative. The primary objectives of RADx-UP are to understand and mitigate the disparities in COVID-19-related morbidity and mortality experienced by communities disproportionately impacted by the COVID-19 pandemic across the United States (https://radx-up.org/). Cross-sectional baseline data from the parent study are used in this analysis.

### Study setting and participants

Data were collected across North Carolina, a state with a population of over 1 million Latines, over a third of whom are foreign-born [2, 22]. To recruit participants, we relied on our partnership's extensive community networks throughout North Carolina, utilizing recruitment flyers and word-of-mouth strategies. To be eligible for this study, participants (a) self-identified as Hispanic/Latine, (b) spoke Spanish fluently, (c) were ≥18 years old, and (d) provided informed consent. Between February 9 and March 11, 2023, we recruited, enrolled, and collected baseline data from 20 Navegantes and their social network members (n = 8 unique social network members per Navegante) for a total of 180 Spanish-speaking Latines.

Data were collected via an interviewer-administered structured individual assessment and entered using REDCap (Research Electronic Data Capture). Data were collected on the phone by trained native Spanish-speaking interviewers to assist with challenges associated with low-literacy rates and/or poor vision. Each assessment was completed in about 45 minutes, depending on responses to specific items and corresponding branching logic.

### Ethics statement

Human subject oversight was provided by the Institutional Review Board of Wake Forest University School of Medicine. Written informed consent was obtained from each participant.

### Measurement

The structured individual assessment contained a total of 142 items with predefined response categories. Most items were drawn from the RADx-UP Common Data Elements and PhenX toolkit (https://www.phenxtoolkit.org/) to ensure standardization of measurement across studies funded by the RADx-UP initiative. Our partnership supplemented these items with additional items that we have used within Latine communities in the US South. The assessment was organized by domain and included sociodemographics, insurance status, immigration documentation, behaviors, testing status and barriers, vaccination status and barriers, and psychosocial factors.

All measures have been validated among Spanish-speaking Latine populations living in the United States previously.

### Dependent variables

Ever testing for COVID-19 was assessed using the item: "Have you ever been tested for COVID-19?". Ever vaccination for COVID-19 and whether vaccination was up to date was assessed using the items: "Have you received a COVID-19 vaccination?", "How many shots against COVID-19 have you received?", "Who was the manufacturer of the first shot you received?", "Who was the manufacturer of the most recent shot you received?", "On what date did you receive your first shot against COVID-19?", and "On what date did you receive your most recent shot against COVID-19?". Participants were considered up to date if they had received an appropriate primary series (for mRNA vaccines, 2 doses, and for adenoviral vector vaccines, 1 dose) and had received a booster with a bivalent formulation in September 2022 or later; this criterion was based on the recommendations for vaccination at the time of assessment completion [23]. These items were required by the RADx-UP initiative or adapted from required items.

### Independent variables

Sociodemographic data included age in years, gender identity, country of origin, language fluency, years living in North Carolina, educational attainment, employment status, and household income. Insurance status was assessed using the item: "What is the primary kind of health insurance or health care plan that you have now?" with response options including: "I do not have health instance", "Private (purchased directly or through employment)", "Public (Medicare, Medicaid, Tricare)", "Don't know", and "Prefer not to answer". These items were required by the RADx-UP initiative or adapted from required items. Self-reported general health status was assessed using the item: "Would you say your health in general is excellent (1), very good (2), good (3), fair (4), or poor (5)?" This commonly used item was also required

by the RADX-UP initiative and is highly reliable and considered to be a good assessment of a person's well-being [24].

Immigration documentation status was assessed using the following three items: "Do you currently have a "green card?", "Do you currently have a valid student visa, valid tourist visa, work permit, or another legal immigration status such as deferred action?", and "Do you have or have you obtained citizenship in the US?" response options included "No", "Yes", and "Prefer not to answer". These items have been validated previously [14].

Eight barriers to testing and 10 barriers to vaccination were assessed. Barriers to getting tested were: "Need to take time off work to get tested"; "out-of-pocket costs for test"; "out-of-pocket costs for transportation, childcare, or time off work"; "I do not know where to go to be tested"; "pain or discomfort from the test"; "saliva"; "concern about others handling my personal data"; and "other". Barriers to getting vaccinated were: "I'm allergic to vaccines"; "I don't like needles"; "I'm not concerned about getting really sick form COVID-19"; "I'm concerned about side effects from the vaccine"; "I don't think vaccines work very well"; "I don't trust that the vaccine will be safe"; "I don't believe the COVID-19 pandemic is as bad as some people say it is"; "I don't want to pay for it"; I don't know enough about how well a COVID-19 vaccine works"; and "other". These items were required by the RADx-UP initiative.

We also assessed social support using the Oslo-3 Social Support Scale [25]. The sum score of the three items ranges from 3 to 14, with high values representing strong levels and low values representing poor levels of social support; this score can be operationalized into three broad categories of social support: (1) 3–8 poor social support, (2) 9–11 moderate social support, (3) 12–14 strong social support.

We assessed COVID-19 testing intention using the single item "I plan to get tested as often as needed"; this item was required by the RADx-UP initiative. Provider communication self-efficacy was assessed using a single item from the Patient Activation Measure [26]: "I am confident I can tell my health care provider concerns I have even when they do not ask". Provider discrimination was assessed using two items developed by our CBPR partnership: "Health care providers discriminate against people like me" and "I have personally been treated unfairly by doctors or healthcare workers because of my ethnicity or race". Testing intention, provider communication self-efficacy, and discrimination were scored on a Likert-scale from Strongly disagree (1) to Strongly agree (4).

Latine community attachment was assessed using the item: "How much do you feel a part of or connected to the Latino community?" Response options ranged from "Not at all a part of or connected to" (0) to "Very much a part of or connected to" (3). This item was developed by our CBPR partnership and has been successfully used in the past [27, 28].

## Data analysis

Descriptive statistics such as mean (*M*), standard deviation (*SD*), frequency (*n*), and percentage (%) were used to describe the sample and study measures. Coefficient omega was used to estimate reliability via internal consistency for social support. Given sample size and model building recommendations [29], we first performed univariate screening of potential independent variables (to use in multivariable modeling) using Akaike information criteria (AIC) [30] and *c* statistics. AIC is useful for comparing models in a relative sense between two or more candidate regression models. The AIC calculation is a function of the number of independent variables used to build the model and the estimate of the likelihood for the model. The lower the AIC score the better. The number of independent variables used per outcome (i.e., COVID-19 testing and vaccination) for each multivariable model was selected based on established guidelines [29].

After the selection of the independent variables (based on AIC), multivariable logistic regression was performed to model relationships between independent variables and ever testing, ever vaccination, and up-to-date vaccination. Adjusted odds ratios (AORs) and their 95% confidence intervals (CIs) were estimated for model effects. Similarly, linear regression was performed for agreement rating of getting tested as needed. A random intercept for the participant social network was specified to account for potential within-network clustering, which was estimated using the intraclass correlation coefficient (ICC). Multicollinearity was assessed with variance inflation factors (VIFs), and all VIFs were less than three except for years in the United States and North Carolina; thus, these were modeled separately where appropriate. A two-sided $p$-value $< 0.05$ was considered statistically significant.

All analyses were performed in SAS v9.4 (SAS Institute, Cary, NC).

## Results

### Sample description

Among the total sample of participants (N = 180), the mean age was 41.7 years (*SD* = 12.9), and 77.2% identified as cisgender women. Most were born outside of the United States; the most frequent Latin American countries of origin included Mexico (43.9%), Colombia (12.8%), and El Salvador (6.8%). The majority of participants (87.2%) had resided in the United States for more than two years, and the mean number of years living in North Carolina was 14.1 years (*SD* = 8.7; range: 0.2–36.0). Many participants also reported educational attainment of less than high school or GED equivalent (24.4%) or high school or GED equivalent (36.7%), currently working (67.2%), lacking health insurance (66.1%), and lacking US immigration documentation (51.1%).

### Barriers to COVID-19 testing and vaccination

Nearly half (46.7%) of participants reported experiencing one or more barriers to COVID-19 testing, and more than one-quarter (28.3%) reported experiencing barriers to COVID-19 vaccination. The mean number of barriers for testing and vaccination were 0.6 (*SD* = 0.7) and 0.5 (*SD* = 0.9), respectively.

### Other psychosocial variables

Participants reported moderate levels of testing intentions with a mean of 2.6 (*SD* = 0.9), overall social support with a mean score of 10.1 (*SD* = 2.0), provider communication self-efficacy with a mean score of 2.8 (*SD* = 0.4), provider discrimination with a mean score of 2.4 (*SD* = 0.5), and Latine community attachment with a mean score of 1.8 (*SD* = 0.9). Table 1 outlines selected study sample characteristics.

### COVID-19 testing and COVID-19 vaccination

The vast majority of the sample reported ever being tested (83.3%; 95% CI = 77.2%, 88.1%) and ever being vaccinated (84.4%; 95% CI = 78.4%, 89.0%). This includes at least one dose of the recommended vaccines. However, only 24.4% were up to date on their vaccination (95% CI = 18.7%, 31.2%). Of the 136 participants who were not up to date with their vaccination, 79.4% (*n* = 108) reported having at least one COVID-19 vaccine.

### Logistic regression modeling

Univariate analyses identified being documented (AIC = 146.0), educational attainment (AIC = 148.3), provider discrimination (AIC = 154.2), and time in the United States

**Table 1. Sample characteristics of the participants (*N* = 180).**

| Characteristic | *n* (%) or *M* (*SD; Min-Max*) |
|---|---|
| Age (years) | 41.7 (12.9; 19–90) |
| Gender identity | |
| Woman | 139 (77.2) |
| Man | 39 (21.7) |
| Transgender woman/Male-to-female (MTF) | 1 (<1.0) |
| Agender | 1 (<1.0) |
| Country of origin | |
| México | 79 (43.9) |
| Colombia | 23 (12.8) |
| El Salvador | 12 (6.8) |
| USA/Puerto Rico | 15 (8.3) |
| Other Latin American countries | 50 (27.8) |
| Missing | 1 (<1.0) |
| Language fluency | |
| Only Spanish | 66 (36.7) |
| More Spanish than English | 82 (45.6) |
| Both Equally | 30 (16.7) |
| More English than Spanish | 2 (1.1) |
| Years living in the United States | 15.6 (9.5; 0.2–37.0) |
| Years living in North Carolina | 14.1 (8.7; 0.2–36.0) |
| Educational attainment | |
| Less than high school or GED equivalent | 44 (24.4) |
| High school or GED equivalent | 66 (36.7) |
| Some college level/technical/vocational degree | 47 (26.1) |
| Bachelor's degree/Other advanced degrees | 23 (12.8) |
| Employment | |
| Currently working | 121 (67.2) |
| Keeping house | 22 (12.2) |
| Student | 2 (1.1) |
| Temporarily laid off, on sick leave, or parental leave | 9 (5.0) |
| Unemployed | 19 (10.6) |
| Retired | 6 (3.3) |
| Disabled, permanently or temporarily | 1 (<1.0) |
| Household income before taxes | |
| < $15,000 | 25 (13.9) |
| $15,000 - $19,999 | 25 (13.9) |
| $20,000 - $24,999 | 8 (4.4) |
| $25,000 - $34,999 | 18 (10.0) |
| $35,000 - $49,999 | 6 (3.3) |
| $50,000 - $74,999 | 7 (3.9) |
| $75,000 - $99,999 | 4 (2.2) |
| Prefer not to answer | 87 (48.3) |
| General health status | 3.1 (0.9; 1–5) |
| Health insurance coverage | |
| None | 119 (66.1) |
| Private | 41 (22.8) |
| Public (Medicare, Medicaid, Tricare) | 20 (11.1) |

(*Continued*)

**Table 1.** (Continued)

| Characteristic | n (%) or M (SD; Min-Max) |
|---|---|
| Immigration documentation status | |
| Currently have a green card | 17 (9.4) |
| Have a valid visa, work permit, or other legal immigration status | 36 (20.0) |
| Citizenship in the US | 27 (15.0) |
| Barriers to getting tested (select all that apply) | |
| Need to take time off work to get tested | 21 (11.7) |
| Out of pocket costs for test | 21 (11.7) |
| Out of pocket costs for transportation, childcare, or time off work | 7 (3.9) |
| I do not know where to go to be tested | 23 (12.8) |
| Pain or discomfort from the test | 12 (6.8) |
| Concern about others handling my personal data | 5 (2.8) |
| Other | 15 (8.3) |
| None | 94 (52.2) |
| Don't know | 2 (1.1) |
| Mean number of barriers | 0.6 (0.7; 0–5) |
| Any barriers | 84 (46.7) |
| Barriers to getting vaccinated (select all that apply) | |
| I'm allergic to vaccines. | 0 |
| I don't like needles. | 7 (3.9) |
| I'm not concerned about getting really sick form COVID-19. | 7 (3.9) |
| I'm concerned about side effects from the vaccine. | 26 (14.4) |
| I don't think vaccines work very well. | 16 (8.9) |
| I don't trust that the vaccine will be safe. | 11 (6.1) |
| I don't believe the COVID-19 pandemic is as bad as some people say it is. | 4 (2.2) |
| I don't want to pay for it. | 0 |
| I don't know enough about how well a COVID-19 vaccine works. | 7 (3.9) |
| Other | 3 (1.7) |
| None | 129 (71.7) |
| Number of barriers | 0.5 (0.9; 0–6) |
| Any barriers | 51 (28.3) |
| Testing intentions | 2.6 (0.9; 0–4) |
| Social support: Oslo-3 | 10.1 (2.0; 5–14) |
| Poor (score 3–8) | 33 (18.3) |
| Moderate (score 9–11) | 99 (55.0) |
| Strong (score 12–14) | 46 (25.6) |
| Missing | 2 (1.1) |
| Provider communication self-efficacy | 2.8 (0.4; 1–4) |
| Provider discrimination | 2.4 (0.5; 1–4) |
| Latine community attachment | 1.8 (0.9; 0–3) |

(AIC = 155.3) as most relevant for ever been COVID-19 tested (max AIC = 159.6). In multi-variable modeling, less than high school/GED equivalent (AOR = .12, 95% CI = .02, .57, $p$ = .008) and high school/GED equivalent (AOR = .19, 95% CI = .04, .86] $p$ = .03) versus more than high school/GED equivalent were significantly associated with lower odds of ever testing (Table 2).

Univariate analyses identified having any vaccination barriers (AIC = 84.4), being documented (AIC = 136.0), being born in US (AIC = 144.1), and general health status

**Table 2. Multivariable logistic regression modeling: Ever tested for COVID-19.**

| Characteristic | AOR | 95% CI | *P*-value |
|---|---|---|---|
| Years in US | 0.97 | 0.92, 1.02 | .24 |
| Documented (Yes vs. No) | 1.46 | 0.50, 4.22 | .48 |
| Education | | | |
| **Less than high school/GED equivalent** | **0.12** | **.02, .57** | **.008** |
| **High school/GED equivalent** | **0.19** | **.04, .86** | **.03** |
| More than High school graduate/GED^RC | - | | |
| Any testing barriers | 0.46 | .17, 1.2 | .11 |
| Perceived provider discrimination | 0.69 | .25, 1.88 | .47 |

*Notes*: Modeling adjusts for social network clustering. Bold = Statistically significant. RC = Reference category.

(AIC = 145.7) as most relevant for ever been COVID-19 vaccination (max AIC = 154.0). In multivariable modeling, having any vaccination barriers (AOR = .006, 95% CI = .001, .059, *p* < .001) was significantly associated with lower odds of ever vaccination (Table 3).

Univariate analyses for being up to date on vaccination identified having any vaccination barriers (AIC = 164.8), general health status (AIC = 166.7), having any testing barriers (AIC = 166.8), being documented (AIC = 167.5), and being born in United States (AIC = 168.2) as most relevant for multivariable modeling (max AIC = 173.6). In multivariable modeling, no univariately screened characteristics were statistically significantly associated with being up to date on vaccination in multivariable modeling (Table 4).

## Discussion

This analysis aimed to explore COVID-19 testing and vaccination within a sample of Spanish-speaking Latines in North Carolina; over half of our sample lacked US immigration documentation. Throughout the pandemic, this population has experienced disproportionate rates of COVID-19-related morbidity and mortality, and Latines currently remain a highly vulnerable population. Data used in this analysis are part of a parent study designed to understand COVID-19 testing and vaccination and develop, implement, and evaluate an intervention designed to promote COVID-19 testing and vaccination among Latine communities [1].

This research is critical given that data were collected in 2023, well into the COVID-19 pandemic; COVID-19 testing became widespread in Spring 2020, and COVID-19 vaccination began in December 2020. Participants had high rates of both testing and vaccination, suggesting a willingness within this community to engage with public health efforts designed to control the spread of COVID-19. However, participants with lower educational attainment (i.e., high school or GED equivalent or below) were less likely to report ever testing. Our findings

**Table 3. Multivariable logistic regression modeling: Ever vaccinated for COVID-19.**

| Characteristic | AOR | 95% CI | *P*-value |
|---|---|---|---|
| Born in US | .23 | .005, 9.3 | .43 |
| Years in North Carolina | .98 | .89, 1.07 | .58 |
| General Health | 2.08 | .84, 5.16 | .11 |
| Documented (Yes vs. No) | 4.39 | .42, 45.7 | .21 |
| **Any vaccination barriers** | **0.006** | **.001, .059** | **< .001** |

*Notes*: Modeling adjusts for social network clustering. Bold = Statistically significant.

**Table 4. Multivariable logistic regression modeling: Up to date on COVID-19 vaccination.**

| Characteristic | AOR | 95% CI | P-value |
|---|---|---|---|
| General Health | 1.45 | .89, 2.36 | .14 |
| Documented (Yes vs. No) | .96 | .42, 2.2 | .92 |
| Any testing barriers | .54 | .22, 1.31 | .17 |
| Any vaccination barriers | .40 | .13, 1.28 | .12 |
| Oslo-3 social support | 1.06 | .86, 1.32 | .59 |

*Notes*: Modeling adjusts for social network clustering. Bold = Statistically significant.

align with existing literature, which suggests that educational disparities can significantly impact health behaviors [1, 7]. Thus, future efforts to increase testing could benefit from tailored strategies that are designed to reach those with lower educational attainment with messages that are both understandable and meaningful.

Furthermore, participants with any vaccination barriers were less likely to report being ever vaccinated. The most commonly reported vaccine barriers included: concerns about the side effects of the vaccine, lack of confidence in vaccine effectiveness, and worries about vaccine safety. Our findings are consistent with research indicating that concerns about side effects can hinder vaccination efforts [8–12]. These concerns may stem from misinformation or previous negative experiences with the US healthcare system. Future efforts to increase vaccination could benefit from tailored strategies and messages that are designed to address these concerns.

Moreover, despite COVID-19 vaccine availability, less than a quarter of participants were up to date with vaccination. This means that while a significant number initiated vaccination, a considerably smaller percentage remained current. Among the 136 participants who were not up to date on their vaccination, 79.4% had received at least one COVID-19 vaccine. Our study contributes to the existing literature by demonstrating that participants who had been initially vaccinated had experiences with vaccination and public health systems but did not reengage as needed. This suggests that their acceptance of vaccines and testing may have been driven by specific circumstances, such as emergency conditions or work and travel requirements rather than the awareness of the importance of testing and vaccination. Given those barriers to testing and vaccination were low, reasons for not being up to date could have included COVID-19 fatigue, confusing messaging about who needed vaccination (including boosters), potential negative side effects, and/or perceptions that COVID-19 was less serious over time. Because these participants would have been identifiable through their vaccination records, there may have been a missed opportunity for targeted, tailored, and personalized outreach about updating one's COVID-19 to ensure higher rates of being up to date.

In this analysis, over half of participants were undocumented, and we expected immigration documentation status to be associated with COVID-19 testing, vaccination, and being up to date with vaccinations. It is well established that being undocumented can be a profound barrier to health care [31]. However, in our modeling, we did not find an association. This may be because Latine communities found alternative ways to access COVID-19 testing and vaccination, such as community health centers, mobile clinics, or community-wide public health testing and vaccination campaigns, which have been, for the most part, less contingent on documentation status throughout the COVID-19 pandemic than other types of healthcare providers. Furthermore, during the height of the pandemic, testing and vaccination were sometimes paired with other services, such as food pantries, and in places that were familiar to Latines [7]. Additionally, government mandates or public health initiatives may have

prioritized equitable access to COVID-19 services regardless of immigration status, thereby mitigating the impact of documentation status as a barrier.

## Limitations

Several limitations to the study should be acknowledged. First, these findings are based on cross-sectional data. Additional studies using a prospective cohort design will be necessary to evaluate the significance and stability of these findings over time. Furthermore, the results of this study may not apply to the general population of Latine adults. However, the degree of fit between a sample and a target population about which generalizations can be made is a common challenge in many studies; in fact, many studies of COVID-19 testing and vaccination among Latine communities are based on non-random and potentially nonrepresentative samples. In this case, however, we have reached a highly hidden, underserved, and vulnerable population: Spanish-speaking and immigrant Latines in North Carolina. Furthermore, this structured individual assessment was interviewer administered, which in some populations can lead to socially desirable responses. However, our CBPR partnership has learned that careful rapport building can result in increased understanding of items and more accurate responses. Additionally, the interviewer used validated wording that emphasized confidentiality as well as the importance of honest answers.

## Strengths

This study provides several significant strengths that enhance its contribution to the understanding of COVID-19 uptake in the Spanish-speaking Latine community. First, by engaging community leaders (Navegantes), we effectively reached a hard-to-access population. These trusted leaders facilitated participation, fostering a sense of safety and encouraging community members. Also, our research team included Spanish-speaking Latines who were not only fluent in the language but also familiar with the cultural nuances of the community. This allowed us to communicate effectively be culturally congruent, which are essential for building trust and rapport and ensuring comprehension. The team also had the flexibility to talk with participants at times convenient for them, accommodating their schedules and increasing the likelihood of participation. This adaptability is crucial when working with communities that may have varying commitments and availability.

## Conclusions

Spanish-speaking Latines face challenges in getting tested and vaccinated for COVID-19. Although ever testing and ever vaccination rates were high in this study, being up to date with recommended vaccinations was low. Educational attainment and concerns about the vaccine were associated with COVID-19 testing and vaccination, respectively. To promote health equity, our findings suggest the need for culturally congruent strategies to address the barriers facing Spanish-speaking Latines and immigrant Latines in the United States.

## Author Contributions

**Conceptualization:** Sandy K. Aguilar-Palma, Thomas P. McCoy, Amanda E. Tanner, Mark A. Hall, Scott D. Rhodes.

**Data curation:** Sandy K. Aguilar-Palma, Thomas P. McCoy, Lilli Mann-Jackson, Jorge Alonzo, Mohammed Sheikh Eldin Jibriel, Dorcas Mabiala Johnson, Tony Locklear, Laurie P. Russell, Scott D. Rhodes.

**Formal analysis:** Sandy K. Aguilar-Palma, Thomas P. McCoy, Mohammed Sheikh Eldin Jibriel, Dorcas Mabiala Johnson, Tony Locklear, Amanda E. Tanner, Alain G. Bertoni, Scott D. Rhodes.

**Funding acquisition:** Amanda E. Tanner, Scott D. Rhodes.

**Methodology:** Sandy K. Aguilar-Palma, Thomas P. McCoy, Scott D. Rhodes.

**Project administration:** Sandy K. Aguilar-Palma, Scott D. Rhodes.

**Supervision:** Amanda E. Tanner, Scott D. Rhodes.

**Writing – original draft:** Sandy K. Aguilar-Palma, Thomas P. McCoy, Mohammed Sheikh Eldin Jibriel, Dorcas Mabiala Johnson, Tony Locklear, Scott D. Rhodes.

**Writing – review & editing:** Lilli Mann-Jackson, Jorge Alonzo, Amanda E. Tanner, Mark A. Hall, Alain G. Bertoni, Ana D. Sucaldito, Laurie P. Russell, Scott D. Rhodes.

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
