## [Decision Letter · Decision Letter 0]

4 Sep 2024

PONE-D-24-29547Characteristics Influencing COVID-19 Testing and Vaccination among Spanish-Speaking Immigrant Latine Persons in North CarolinaPLOS ONE

Dear Dr. Aguilar-Palma,

Thank you for submitting your manuscript to PLOS ONE. After careful consideration, we feel that it has merit but does not fully meet PLOS ONE’s publication criteria as it currently stands. Therefore, we invite you to submit a revised version of the manuscript that addresses the points raised during the review process.

Your manuscript is generally well-written; however, it needs improvements to fully meet the publication criteria required by PLOS ONE. Please address the following issue in your revised manuscript:**"Experiments, statistics, and other analyses are performed to a high technical standard and are described in sufficient detail."**

We look forward to receiving your revised manuscript.

Kind regards,

Weijun Yu, Ph.D., M.D., M.S.

Academic Editor

PLOS ONE

https://journals.plos.org/plosone/article?id=10.1371%2Fjournal.pone.0296812

https://bmjopen.bmj.com/content/12/11/e066585.full

In your revision ensure you cite all your sources (including your own works), and quote or rephrase any duplicated text outside the methods section. Further consideration is dependent on these concerns being addressed.

“This work is supported by the US National Institute of Minority Health and Health Disparities [U01MD017431; PIs: Scott D. Rhodes and Amanda E. Tanner].”

“This work is supported by the US National Institute of Minority Health and Health Disparities [https://www.nimhd.nih.gov/; U01MD017431; PIs: SDR and AET]. The funders had no role in study design, data collection and analysis, decision to publish, or preparation of the manuscript.”

4. For studies involving third-party data, we encourage authors to share any data specific to their analyses that they can legally distribute. PLOS recognizes, however, that authors may be using third-party data they do not have the rights to share. When third-party data cannot be publicly shared, authors must provide all information necessary for interested researchers to apply to gain access to the data. (https://journals.plos.org/plosone/s/data-availability#loc-acceptable-data-access-restrictions)

a) A description of the data set and the third-party source

b) If applicable, verification of permission to use the data set

c) Confirmation of whether the authors received any special privileges in accessing the data that other researchers would not have

d) All necessary contact information others would need to apply to gain access to the data

Reviewers' comments:

Reviewer's Responses to Questions

**Comments to the Author**

1. Is the manuscript technically sound, and do the data support the conclusions?

Reviewer #1: Yes

Reviewer #2: Yes

2. Has the statistical analysis been performed appropriately and rigorously? 

Reviewer #1: Yes

Reviewer #2: Yes

3. Have the authors made all data underlying the findings in their manuscript fully available?

Reviewer #1: Yes

Reviewer #2: Yes

4. Is the manuscript presented in an intelligible fashion and written in standard English?

Reviewer #1: Yes

Reviewer #2: Yes

5. Review Comments to the Author

Reviewer #1: Comment to the authors:

At the end of the subheading, COVID-19 and Latine Communities in the United States, you mentioned, “Throughout the pandemic and continuing today, these disparities have been notably more pronounced among Spanish speakers and Spanish-speaking immigrants.”

Please specify what you mean by Spanish speakers. Are you referring to a first, second or third generation of US-born latines or undocumented latines? Be precise.

What study you are referring to has been clarified in the paragraph below. Are you referring to the preliminary research you conducted before this study? Please specify that.

“Participants in this study reported that many Latines do not trust, are discouraged from using, and face difficulty accessing and navigating the U.S. healthcare system, particularly for non-English speakers and those less familiar with accessing healthcare resources and services in the United States. Participants in this study reported that many Latines do not trust, are discouraged from using, and face difficulty accessing and navigating the U.S. healthcare system, particularly for non-English speakers and those less familiar with accessing healthcare resources and services in the United States.”

Methodology section:

Study Design:

1. Please provide information about the study design used.

Instrumentation:

1. Please describe how the instrument tool used to collect data was organized.

2. Did you develop the instrument tool? Please provide this information in the manuscript.

3. You said that coefficient omega was used to estimate reliability via internal consistency for social support. What about the other items in the instrument tool? Did you estimate their reliability as well?

4. Did you assess the validity of the instrument tool?

5. If the instrument has been developed by someone else, please specify the type of validity and reliability conducted by the authors who created the instrument tool.

6. If you did not perform the validity and reliability of the instrument tool, please include these in the limitation section of your study.

Participants

You said that you used 20 “navegantes” and their social networks to recruit participants.

1. How many times did each “navegante” have to contact potential participants?

2. Did you follow a specific methodology for contacting participants, such as the modified tailored design method (Dillan, 2014)? Please describe the methodology followed.

Independent Variables:

You said: “Eight barriers to testing and 10 barriers to vaccination were assessed, including taking time off from work, out-of-pocket costs, not knowing where to go, pain or discomfort, and worries about confidentiality. Please name all the barriers to testing and vaccination.

Data Analysis:

1. How did you analyze the data? Did you use a software such as SPSS, JMP, R, or Excel? If so, please describe the software utilized for data analysis and the version.

2. For those unfamiliar with the Akaike information criterion, please briefly describe what it is used for, explain the best-fit model you found, and state the AIC weight of the model.

You presented the results of your data clearly.

Discussion

The author engages in three steps in a discussion section: interpretation, analysis, and explanation. An adequate discussion section will also explain why the research results are essential and where they fit in the current literature while being self-critical and honest about the study's shortcomings. In your discussion, you must mention how your data fits the literature. The results from your research are similar/different to those found by other authors? Also, you must make more explicit about the novelty that your study found and what your contributions to the literature are.

Overall, you did an a very good job!

Reviewer #2: Thank you for the opportunity to review this paper. Overall, this is a well-written paper. I have few comments detailed below.

Overall: The title and most of the paper indicates that the research was conducted among immigrant Latine population, however, upon reviewing this paper in entirety, it does not seem to be true. The authors have included all Latine population of the specified geographic region, irrespective of their immigration status. The authors should consider clarifying this throughout the manuscript.

Abstract: The authors mention challenges to COVID testing/vaccinations in their conclusions but do not report the challenges in the results. The authors should consider updating the results with the said challenges.

Introduction: While the authors focus their research question on Latine immigrants, they report the COVID vaccinations among overall Latine population and is not specific to Latine immigrants. The authors should consider clarifying the population and update the introduction to report existing literature and the research gap based on that.

Methods: It seems like authors have assessed all Latinas in NC and not just immigrants. This information is confusing based on the title and abstract that focuses on immigrants only. The authors should consider clarifying if overall Latine population of NC constituted the study cohort or just Latine immigrant population.

Discussion: The discussion currently is only an extension of the results where the authors explained and interpreted the findings. The authors should consider updating the discussion with comparison of the findings related to COVID testing/vaccination, challenge, barriers etc, with other races (non-Hispanic White/ Black) in NC/United States. Else, it is hard to understand whether the testing/vaccinations rates, or challenges are indeed lower/higher/comparable to the other races, and if any interventions are required for this specific population.

The authors should exercise caution with conclusions especially since the sample size is very small.

One of the limitations states that the findings cannot be generalized to the overall Latine population, would the authors clarify the study cohort throughout this manuscript, whether they are immigrant Latine population or overall Latine population of NC

The authors should consider adding strengths of this study

6. PLOS authors have the option to publish the peer review history of their article (what does this mean?). If published, this will include your full peer review and any attached files.

Reviewer #1: **Yes: **Dr. Maria Mercedes Rossi

Reviewer #2: **Yes: **Dr. Rajrupa Ghosh

---

## [Author Response · Author response to Decision Letter 0]

13 Dec 2024

Reviewers' comments:

Reviewer #1: 

• At the end of the subheading, COVID-19 and Latine Communities in the United States, you mentioned, “Throughout the pandemic and continuing today, these disparities have been notably more pronounced among Spanish speakers and Spanish-speaking immigrants.”

Please specify what you mean by Spanish speakers. Are you referring to a first, second or third generation of US-born latines or undocumented latines? Be precise.

Response: Thank you for pointing this out. Our study included Spanish speakers of Latin origin from all immigration statuses, including those born in their country of origin and those born in the United States. We do not have data on whether participants are first, second, or third generation. To simplify our description, we will use the term "Spanish-speaking Latines" and eliminate "Spanish-speaking immigrants." 

• What study you are referring to has been clarified in the paragraph below. Are you referring to the preliminary research you conducted before this study? Please specify that.

“Participants in this study reported that many Latines do not trust, are discouraged from using, and face difficulty accessing and navigating the U.S. healthcare system, particularly for non-English speakers and those less familiar with accessing healthcare resources and services in the United States. Participants in this study reported that many Latines do not trust, are discouraged from using, and face difficulty accessing and navigating the U.S. healthcare system, particularly for non-English speakers and those less familiar with accessing healthcare resources and services in the United States.”

Response: We appreciate your bringing this to our attention. This paragraph refers to our community-based participatory research (CBPR) partnership’s preliminary research. We are clarifying and incorporating “preliminary research” instate “study” into the paragraph. We also added a citation for this preliminary research.

• Methodology section: 

Study Design:

1. Please provide information about the study design used.

Response: We appreciate pointing this out. We added to the Methods section: Cross-sectional baseline data from the parent study are used in this analysis.

Instrumentation:

1. Please describe how the instrument tool used to collect data was organized.

Response: Thank you for your comment. We added to the Measurement section: “The assessment was organized by domain and included sociodemographics, insurance status, immigration documentation, behaviors, testing status and barriers, vaccination status and barriers, and psychosocial factors.”

2. Did you develop the instrument tool? Please provide this information in the manuscript.

Response: Thank you for pointing this out. We added to Measurement section: Most items were drawn from the RADx-UP Common Data Elements and PhenX toolkit (https://www.phenxtoolkit.org/) to ensure standardization of measurement across studies funded by the RADx-UP initiative. Our partnership supplemented these items with additional items that we have used within Latine communities in the US South.

3. You said that coefficient omega was used to estimate reliability via internal consistency for social support. What about the other items in the instrument tool? Did you estimate their reliability as well?

Response: Thank you for your comment. Social support has 3 items; it is the only scale in this analysis for which omega use is appropriate. 

4. Did you assess the validity of the instrument tool?

Response: Thank you for your question. Yes, as we noted, “All measures have been validated among Spanish-speaking Latine populations living in the United States previously.”

5. If the instrument has been developed by someone else, please specify the type of validity and reliability conducted by the authors who created the instrument tool.

Response: Thank you for your comment. We have detailed in the Dependent variables and Independent variables sections where items on the assessment originated. 

6. If you did not perform the validity and reliability of the instrument tool, please include these in the limitation section of your study.

Response: Thank you for your comment. We have detailed in the Dependent variables and Independent variables sections where items on the assessment originated.

• Participants

You said that you used 20 “navegantes” and their social networks to recruit participants.

1. How many times did each “navegante” have to contact potential participants?

Response: Thank you for pointing this out. Regarding recruitment, we do not have a record of how many times each Navegante contacted potential social network participants. The Navegantes provided us with their lists of eight social network participants only after social network participants had agreed to participate.

2. Did you follow a specific methodology for contacting participants, such as the modified tailored design method (Dillan, 2014)? Please describe the methodology followed.

Response: We appreciate this important question. As we describe in our Study Setting and Participants section, because each Navegante must be a leader within existing social networks, we primarily recruited Navegantes through our partnership’s extensive community networks across North Carolina and through word-of-mouth. Each Navegante was responsible for recruiting eight participants from their social network; they generally did this through in-person communication.

• Independent Variables:

You said: “Eight barriers to testing and 10 barriers to vaccination were assessed, including taking time off from work, out-of-pocket costs, not knowing where to go, pain or discomfort, and worries about confidentiality. Please name all the barriers to testing and vaccination.

Response: Thank you for pointing this out. Below we list the barriers according to our baseline.

Barriers to getting tested:

1. Need to take time off work to get tested

2. Out of pocket costs for test

3. Out of pocket costs for transportation, childcare, or time off work

4. I do not know where to go to be tested

5. Pain or discomfort from the test

6. Saliva

7. Concern about others handling my personal data

8. Other

Barriers to getting vaccinated:

 1. I’m allergic to vaccines

 2. I don’t like needles

 3. I’m not concerned about getting really sick form COVID-19

 4. I’m concerned about side effects from the vaccine

 5. I don’t think vaccines work very well

 6. I don’t trust that the vaccine will be safe

 7. I don’t believe the COVID-19 pandemic is as bas as some people say it is

 8. I don’t want to pay for it

 9. I don’t know enough about how well a COVID-19 vaccine works

 10. Other

We also have included them in the text of the paper.

• Data Analysis: 

1. How did you analyze the data? Did you use a software such as SPSS, JMP, R, or Excel? If so, please describe the software utilized for data analysis and the version.

Response: Thank you for your question. All analyses were performed in SAS v9.4 (SAS Institute, Cary, NC). We are adding this statement to the paper in the data analysis section.

2. For those unfamiliar with the Akaike information criterion, please briefly describe what it is used for, explain the best-fit model you found, and state the AIC weight of the model.

Response: Thank you for your comment. We are adding to the manuscript the following statement: “AIC is useful for comparing models in a relative sense between two or more candidate regression models. The AIC calculation is a function of the number of independent variables used to build the model and the estimate of the likelihood for the model. The lower the AIC score the better.”

• Discussion

The author engages in three steps in a discussion section: interpretation, analysis, and explanation. An adequate discussion section will also explain why the research results are essential and where they fit in the current literature while being self-critical and honest about the study's shortcomings. In your discussion, you must mention how your data fits the literature. The results from your research are similar/different to those found by other authors? Also, you must make more explicit about the novelty that your study found and what your contributions to the literature are.

Response: Thank you for your insightful feedback on the discussion section of our manuscript. We have made revisions to clarify our findings and their significance with existing literature. You can find a revised version of the discussion in the paper that incorporates your suggestions.

Reviewer # 2:

Thank you for the opportunity to review this paper. Overall, this is a well-written paper. I have few comments detailed below.

• Overall: The title and most of the paper indicates that the research was conducted among immigrant Latine population, however, upon reviewing this paper in entirety, it does not seem to be true. The authors have included all Latine population of the specified geographic region, irrespective of their immigration status. The authors should consider clarifying this throughout the manuscript.

Response: Thank you for highlighting this important point. Our study included Spanish speakers of Latino origin from various immigration statuses, although the majority were born in their home countries (89.9%), and later immigrated to the United States. To clarify, we will consistently use the term "Spanish speakers" throughout the manuscript, and in some sections, we specify that they are predominantly immigrants.

• Abstract: The authors mention challenges to COVID testing/vaccinations in their conclusions but do not report the challenges in the results. The authors should consider updating the results with the said challenges.

Response: Thank you for pointing this out. We are updating the results and conclusion in the Abstract to enhance clarity. 

• Introduction: While the authors focus their research question on Latine immigrants, they report the COVID vaccinations among overall Latine population and is not specific to Latine immigrants. The authors should consider clarifying the population and update the introduction to report existing literature and the research gap based on that.

Response: Thank you for highlighting this point. Our study included Spanish speakers of Latino origin from various immigration statuses, although the majority were born in their home countries (89.9%), and later immigrated to the United States. To enhance clarity, we will use the term "Spanish speakers Latines". 

• Methods: It seems like authors have assessed all Latinas in NC and not just immigrants. This information is confusing based on the title and abstract that focuses on immigrants only. The authors should consider clarifying if overall Latine population of NC constituted the study cohort or just Latine immigrant population.

Response: Thank you for emphasizing again this point. We confirm that our study included Spanish speakers of Latino origin from various immigration statuses, although the majority were born in their home countries (89.9%), and later immigrated to the United States. To enhance clarity, we are using just the term "Spanish-speaking Latine" in this section. 

• Discussion: The discussion currently is only an extension of the results where the authors explained and interpreted the findings. The authors should consider updating the discussion with comparison of the findings related to COVID testing/vaccination, challenge, barriers etc, with other races (non-Hispanic White/ Black) in NC/United States. Else, it is hard to understand whether the testing/vaccinations rates, or challenges are indeed lower/higher/comparable to the other races, and if any interventions are required for this specific population.

Response: Thank you for your insightful comment. We agree that conducting a comparative analysis would be very interesting. However, this study was specifically designed and approved to evaluate the behaviors and acceptance of COVID-19 resources among a specific population: Spanish-speaking Latines, rather than comparing them with other racial groups. In this study, we do not have the data necessary to conduct a comparative analysis of different racial groups within the same location and timeframe. We believe your suggestion is excellent and could serve as a valuable avenue for future research.

• The authors should exercise caution with conclusions especially since the sample size is very small.

Response: We appreciate you bringing this to my attention. We have updated our conclusions to make them more precise.

• One of the limitations states that the findings cannot be generalized to the overall Latine population, would the authors clarify the study cohort throughout this manuscript, whether they are immigrant Latine population or overall Latine population of NC.

Response: Thank you for your comment in this section. We confirm that our study included Spanish speakers of Latino origin from various immigration statuses living in North Carolina. To be more specific we are using the term " Spanish-speaking and immigrant Latines in North Carolina" in this section. 

• The authors should consider adding strengths of this study

Response: We appreciate your comment and suggestion. We are adding a strength section to this study. 

We would like to express our gratitude to the reviewers of PLOS ONE for their valuable comments. We have carefully considered each piece of feedback and believe we have addressed all of them. We hope our responses meet your expectations, and we look forward to continuing the process of publishing this relevant article.

---

## [Decision Letter · Decision Letter 1]

6 Jan 2025

Characteristics influencing COVID-19 testing and vaccination among Spanish-speaking Latine persons in North Carolina

PONE-D-24-29547R1

Dear Dr. Aguilar-Palma,

We’re pleased to inform you that your manuscript has been judged scientifically suitable for publication and will be formally accepted for publication once it meets all outstanding technical requirements.

Kind regards,

José Ramos-Castañeda, M.Sc., Ph.D

Academic Editor

PLOS ONE

Additional Editor Comments (optional):

Reviewers' comments:

Reviewer's Responses to Questions

**Comments to the Author**

1. If the authors have adequately addressed your comments raised in a previous round of review and you feel that this manuscript is now acceptable for publication, you may indicate that here to bypass the “Comments to the Author” section, enter your conflict of interest statement in the “Confidential to Editor” section, and submit your "Accept" recommendation.

Reviewer #1: All comments have been addressed

Reviewer #2: All comments have been addressed

2. Is the manuscript technically sound, and do the data support the conclusions?

Reviewer #1: Yes

Reviewer #2: (No Response)

3. Has the statistical analysis been performed appropriately and rigorously? 

Reviewer #1: Yes

Reviewer #2: (No Response)

4. Have the authors made all data underlying the findings in their manuscript fully available?

Reviewer #1: Yes

Reviewer #2: (No Response)

5. Is the manuscript presented in an intelligible fashion and written in standard English?

Reviewer #1: Yes

Reviewer #2: (No Response)

6. Review Comments to the Author

Reviewer #1: The authors have addressed all my comments and suggestions, especially regarding the instrument's validity. I understand better now what survey items were coming from developed survey tools and that the authors have previously validated each instrument and been validated with the Latinx population. Thank you for clarifying that.

With all the changes made, you have a more substantial manuscript. I wish you good luck with your next steps on the project.

Reviewer #2: (No Response)

7. PLOS authors have the option to publish the peer review history of their article (what does this mean?). If published, this will include your full peer review and any attached files.

Reviewer #1: **Yes: **Maria Mercedes Rossi, PhD

Reviewer #2: **Yes: **Rajrupa Ghosh

---

## [Editor Report · Acceptance letter]

11 Jan 2025

PONE-D-24-29547R1 

PLOS ONE

Dear Dr. Aguilar-Palma, 

I'm pleased to inform you that your manuscript has been deemed suitable for publication in PLOS ONE. Congratulations! Your manuscript is now being handed over to our production team.

Kind regards, 

on behalf of

Dr. José Ramos-Castañeda 

Academic Editor

PLOS ONE